# Understanding the Initial Condensation of Convolutional Neural Networks

## Abstract

Previous research has shown that fully-connected neural networks with small initialization and gradient-based training methods exhibit a phenomenon known as condensation during training. This phenomenon refers to the input weights of hidden neurons condensing into isolated orientations during training, revealing an implicit bias towards simple solutions in the parameter space. However, the impact of neural network structure on condensation remains unknown. In this study, we study convolutional neural networks (CNNs) as the starting point to explore the distinctions in the condensation behavior compared to fully-connected neural networks. Theoretically, we firstly demonstrate that under gradient descent (GD) and the small initialization scheme, the convolutional kernels of a two-layer CNN condense towards a specific direction determined by the training samples within a given time period. Subsequently, we conduct a series of systematic experiments to substantiate our theory and confirm condensation in more general settings. These findings contribute to a preliminary understanding of the non-linear training behavior exhibited by CNNs.

## 1 Introduction

As large neural networks continue to demonstrate impressive performance in numerous practical tasks, a key challenge has come to understand the reasons behind the strong generalization capabilities often exhibited by over-parameterized networks (Breiman, 1995; Zhang et al., 2021). A commonly employed approach to understanding neural networks is to examine their implicit biases during the training process. Several studies have shown that neural networks tend to favor simple solutions. For instance, from a Fourier perspective, neural networks have a bias toward low-frequency functions, which is known as the frequency principle (Xu et al., 2019; 2020) or spectral bias (Rahaman et al., 2019). In the parameter space, (Luo et al., 2021) observed a condensation phenomenon, i.e., the input weights of hidden neurons in two-layer ReLU neural networks condense into isolated orientations during training in the non-linear regime, particularly with small initialization. Fig. 1 presents an illustrative example in which a large condensed network can be reduced to an effective smaller network with only two neurons. Based on complexity theory (Bartlett and Mendelson, 2002), as the condensation phenomenon reduces the network complexity, it might provide insights into how over-parameterized neural networks achieve good generalization performance in practice. (Zhang and Xu, 2022) drew inspiration from this phenomenon and found that dropout (Hinton et al., 2012; Srivastava et al., 2014), a commonly used optimization technique for improving generalization, exhibits an implicit bias towards condensation through experiments and theory. Prior literature has predominantly centered on the study of fully-connected neural networks, thereby leaving the emergence and properties of the condensation phenomenon in neural networks with different structural characteristics inadequately understood. Consequently, this paper aims to investigate the occurrence of condensation in convolutional neural networks (CNNs).

The success of deep learning relies heavily on the structures used, such as convolution and attention. Convolution is an ideal starting point for investigating the impact of structure on learning outcomes, as it is widely used and has simple structure. To achieve a clear condensation phenomenon in CNNs, we adopt a strategy of initializing weights with small values. Small weight initialization can result in rich non-linearity of neural network (NN) training behavior (Mei et al., 2019; Rotskoff and Vanden-Eijnden, 2018; Chizat and Bach, 2018; Sirignano and Spiliopoulos, 2020). Over-parameterized NNs

with small initialization can, for instance, achieve low generalization error (Advani et al., 2020) and converge to a solution with maximum margin (Phuong and Lampert, 2020). In contrast to the condensation in fully-connected networks, each kernel in CNNs is considered as a unit, and condensation is referred to the behavior, that a set of kernels in the same layer evolves towards the same direction.

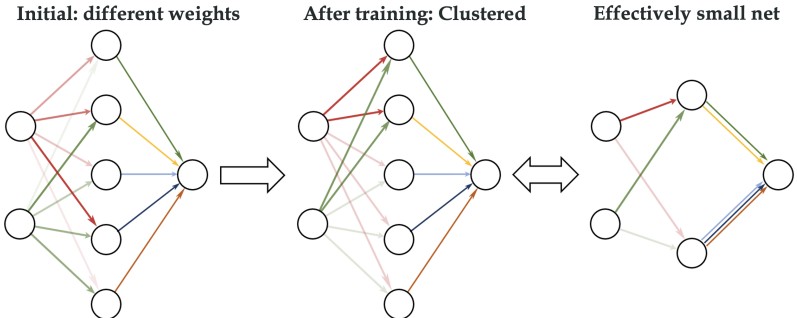

Figure 1: Illustration of condensation. The color and its intensity of a line indicate the strength of the weight. Initially, weights are random. Soon after training, the weights from an input node to all hidden neurons are clustered into two groups, i.e., condensation. Multiple hidden neurons can be replaced by an effective neuron with low complexity, which has the same input weight as original hidden neurons and the same output weight as the summation of all output weights of original hidden neurons.

Understanding the initial condensation can benefit understanding subsequent training stages (Fort et al., 2020; Hu et al., 2020; Luo et al., 2021; Jiang et al., 2019; Li et al., 2018). Previous research has shown how neural networks with small initialization can condense during the initial training stage in fully-connected networks (Maennel et al., 2018; Pellegrini and Biroli, 2020; Lyu et al., 2021; Zhou et al., 2022; Chen et al., 2023; Wang and Ma, 2023). This work aims to demonstrate the initial condensation in CNNs during training. A major advantage of studying the initial training stage of neural networks with small initialization is that the network can be approximated accurately by the leading-order Taylor expansion at zero weights. Further, the structure of CNNs may cause kernels in the same layer to exhibit similar dynamics. Through theoretical proof, we show that CNNs can condense into one or a few directions within a finite training period with small initialization. This initial condensation serves an important role in resetting the neural network of different initializations to a similar and simple state, thus reducing the sensitivity of initialization and facilitating the tuning of hyper-parameters of initialization. Our contribution is summarized as follows:

- Under gradient descent and the small initialization scheme, we demonstrate in theory that the convolutional kernels of two-layer CNNs exhibit the initial condensation phenomenon. This phenomenon reveals that the kernels tend to cluster toward a specific direction determined by the training samples within a given training period.

- Our experimental settings are designed to be more general than the counterparts in the theoretical analysis. Specifically, we demonstrate that kernel weights within the same layer of a three-convolution-layer CNN still tend to cluster together during training when subjected to the small initialization and gradient-based training methods and partly remain till the end. This observation is consistent with the initial condensation phenomenon observed in our theory, and it reinforces our understanding of the non-linear training behavior exhibited by CNNs.

## 2 RELATED WORKS

For fully-connected neural networks, it has been generally studied that different initializations can lead to very different training behavior regimes (Luo et al., 2021), including linear regime (similar to the lazy regime) (Jacot et al., 2018; Arora et al., 2019; Zhang et al., 2020; E et al., 2020; Chizat and Bach, 2019), critical regime (Mei et al., 2019; Rotskoff and Vanden-Eijnden, 2018; Chizat and Bach, 2018; Sirignano and Spiliopoulos, 2020) and condensed regime (non-linear regime). The relative

change of input weights as the width approaches infinity is a critical parameter that distinguishes the different regimes, namely $0$, $O(1)$, and $+\infty$. (Zhou et al., 2022) demonstrated that these regimes also exist for three-convolution-layer ReLU neural networks with infinite width. Experiments suggest that condensation is a frequent occurrence in the non-linear regime(Zhou et al., 2022; Luo et al., 2021).

(Zhang et al., 2021a;b)discovered an embedding principle in loss landscapes between narrow and wide neural networks, based on condensation. This principle suggests that the loss landscape of a deep neural network (DNN) includes all critical points of narrower DNNs, which is also studied in (Fukumizu and Amari, 2000; Fukumizu et al., 2019; Simsek et al., 2021). The embedding structure indicates the existence of global minima where condensation can occur. (Zhang et al., 2022) has demonstrated that NNs exhibiting condensation can achieve the desired function with a substantially lower number of samples compared to the number of parameters. However, these studies fail to demonstrate the training process's role in causing condensation.

CNN is one of the fundamental structures in deep learning (Gu et al., 2018). (He et al., 2016) introduced the use of residual connections for training deep CNNs, which has greatly enhanced the performance of CNNs on complex practical tasks. Recently, there are also many theoretical advances. (Zhou, 2020) shows that CNN can be used to approximate any continuous function to an arbitrary accuracy when the depth of the neural network is large enough. (Arora et al., 2019) exactly compute the neural tangent kernel of CNN. Provided that the signal-to-noise ratio satisfies certain conditions, (Cao et al., 2022) have demonstrated that a two-layer CNN, trained through gradient descent, can obtain negligible training and test losses. In this work, we focus on the training process of CNNs.

## 3 PRELIMINARIES

### 3.1 SOME NOTATIONS

For a matrix $\mathbf{A}$, we use $\mathbf{A}_{i,j}$ to denote its $(i,j)$-th entry. For a high-dimensional tensor, for example, a four-dimensional tensor $\mathbf{W}$, we use $\mathbf{W}_{i,j,k,l}$ to denote its $(i,j,k,l)$-th entry. We also use $\mathbf{W}_{i,j,k,:}$ to denote the $i,j,k$-th row, and so on for other indices. $\mathbf{W}_{:,:,k,l}$ is denoted as the two-dimensional tensor at the $k,l$-th entry.

We let $[n] = \{1, 2, \ldots, n\}$. We set $\mathcal{N}(\boldsymbol{\mu}, \Sigma)$ as the normal distribution with mean $\boldsymbol{\mu}$ and covariance $\Sigma$. We set a special vector $\mathbb{1} := (1, 1, 1, \ldots, 1)^{\mathsf{T}}$, whose dimension varies. For a vector $\mathbf{v}$, we use $\|\mathbf{v}\|_2 \|\mathbf{v}\|_\infty$ to denote its ~~Euclidean~~ maximum norm, and we use $\langle \cdot, \cdot \rangle$ to denote the standard inner product between two vectors. ~~Finally, for a matrix $\mathbf{A}$, we use $\|\mathbf{A}\|_{2 \to 2}$ to denote its operator norm.~~

### 3.2 PROBLEM SETUP

We focus on the empirical risk minimization problem given by the quadratic loss:

$$\min_{\boldsymbol{\theta}} R_S(\boldsymbol{\theta}) = \frac{1}{2n} \sum_{i=1}^{n} \left( f(\boldsymbol{x}_i, \boldsymbol{\theta}) - y_i \right)^2. \tag{1}$$

In the above, $n$ is the total number of training samples, $\{\boldsymbol{x}_i\}_{i=1}^n$ are the training inputs, $\{y_i\}_{i=1}^n$ are the labels, $f(\boldsymbol{x}_i, \boldsymbol{\theta})$ is the prediction function, and $\boldsymbol{\theta}$ are the parameters to be optimized, which is modeled by a $(L+1)$-layer CNN with filter size $m \times m$. We denote $\boldsymbol{x}^{[l]}(i)$ as the output of the $l$-th layer with respect to the $i$-th sample for $l \geq 1$, and $\boldsymbol{x}^{[0]}(i) := \boldsymbol{x}_i$ is the $i$-th training input. For any $l \in [0 : L]$, we denote the size of width, height, channel of $\boldsymbol{x}^{[l]}$ as $W_l$, $H_l$, and $C_l$, respectively, i.e., $\{\boldsymbol{x}^{[l]}(i)\}_{i=1}^n \subset \mathbb{R}^{W_l \times H_l \times C_l}$. We introduce a filter operator $\chi(\cdot, \cdot)$, which maps the width and height indices of the output of all layers to a binary variable, i.e., for a filter of size $m \times m$, the filter operator reads

$$\chi(p, q) = \begin{cases} 1, & \text{for } 0 \leqslant p, q \leqslant m - 1 \\ 0, & \text{otherwise,} \end{cases} \tag{2}$$

then the $(L+1)$-layer CNN with filter size $m \times m$ is recursively defined for $l \in [2:L]$,

$$\boldsymbol{x}_{u,v,\beta}^{[1]} := \left[ \sum_{\alpha=1}^{C_0} \left( \sum_{p=-\infty}^{+\infty} \sum_{q=-\infty}^{+\infty} \boldsymbol{x}_{u+p,v+q,\alpha}^{[0]} \cdot \boldsymbol{W}_{p,q,\alpha,\beta}^{[1]} \cdot \chi(p,q) \right) \right] + \boldsymbol{b}_\beta^{[1]},$$

$$\boldsymbol{x}_{u,v,\beta}^{[l]} := \left[ \sum_{\alpha=1}^{C_{l-1}} \left( \sum_{p=-\infty}^{+\infty} \sum_{q=-\infty}^{\infty} \sigma\left( \boldsymbol{x}_{u+p,v+q,\alpha}^{[l-1]} \right) \cdot \boldsymbol{W}_{p,q,\alpha,\beta}^{[l]} \cdot \chi(p,q) \right) \right] + \boldsymbol{b}_\beta^{[l]},$$

$$f(\boldsymbol{x},\boldsymbol{\theta}) := f_{\mathrm{CNN}}(\boldsymbol{x},\boldsymbol{\theta}) := \left\langle \boldsymbol{a}, \sigma\left( \boldsymbol{x}^{[L]} \right) \right\rangle = \sum_{\beta=1}^{C_L} \sum_{u=1}^{W_L} \sum_{v=1}^{H_L} \boldsymbol{a}_{u,v,\beta} \cdot \sigma\left( \boldsymbol{x}_{u,v,\beta}^{[L]} \right),$$

where $\sigma(\cdot)$ is the activation function applied coordinate-wisely to its input, and for each layer $l \in [L]$, all parameters belonging to this layer are initialized by: For $p,q \in [m-1]$, $\alpha \in [C_{l-1}]$ and $\beta \in [C_l]$,

$$\boldsymbol{W}_{p,q,\alpha,\beta}^{[l]} \sim \mathcal{N}(0,\beta_1^2), \quad \boldsymbol{b}_\beta^{[l]} \sim \mathcal{N}(0,\beta_1^2). \tag{3}$$

Note that for a pair of $\alpha$ and $\beta$, $\boldsymbol{W}_{\cdot,\cdot,\alpha,\beta}^{[l]}$ is called a kernel. Moreover, for $u \in [W_L]$ and $v \in [H_L]$,

$$\boldsymbol{a}_{u,v,\beta} \sim \mathcal{N}(0,\beta_2^2), \tag{4}$$

and for convenience in theory, we set $\beta_1 = \beta_2 = \varepsilon$, where $\varepsilon > 0$ is the scaling parameter.

**Cosine similarity:** The cosine similarity between two vectors $\boldsymbol{u}_1$ and $\boldsymbol{u}_2$ is defined as

$$D(\boldsymbol{u}_1,\boldsymbol{u}_2) = \frac{\boldsymbol{u}_1^\intercal \boldsymbol{u}_2}{(\boldsymbol{u}_1^\intercal \boldsymbol{u}_1)^{1/2}(\boldsymbol{u}_2^\intercal \boldsymbol{u}_2)^{1/2}}. \tag{5}$$

We remark that in order to compute the cosine similarity between two kernels, each kernel $\boldsymbol{W}_{\cdot,\cdot,\alpha,\beta}^{[l]}$ shall be vectorized.

Before we proceed to our theoretical findings, as we consider two-layer CNNs ($L = 1$), the upper case $[l]$ can be omitted since the number of weight vectors is equal to 1, i.e., $\boldsymbol{W}_{\cdot,\cdot,\alpha,\beta} := \boldsymbol{W}_{\cdot,\cdot,\alpha,\beta}^{[1]}$. We denote $M := C_1$, the number of channels in $\boldsymbol{x}^{[1]}(i)$. As for two-layer NNs, with slight misuse of notations, we denote by $\boldsymbol{x}_r^{[1]} := \langle \boldsymbol{w}_r, \boldsymbol{x} \rangle$, then the output function of a two-layer neural networks (NNs) reads

$$f_{\mathrm{TwoLayer}}(\boldsymbol{x},\boldsymbol{\theta}) := \sum_{r=1}^M a_r \sigma(\boldsymbol{x}_r^{[1]}) = \sum_{r=1}^M \left\langle a_r, \sigma(\boldsymbol{x}_r^{[1]}) \right\rangle := \sum_{r=1}^M \left\langle a_r, \sigma(\langle \boldsymbol{w}_r, \boldsymbol{x} \rangle) \right\rangle,$$

and it is noteworthy that the outmost inner product in the above equation is simple multiplication or is with dimension 1, i.e., $\langle \cdot, \cdot \rangle : \mathbb{R} \times \mathbb{R} \to \mathbb{R}$.. By comparison, also with slight misuse of notations, as we denote $\boldsymbol{a}_\beta := \mathrm{vec}(\boldsymbol{a}_{u,v,\beta})$ and $\boldsymbol{x}_\beta^{[1]} := \mathrm{vec}(\boldsymbol{x}_{u,v,\beta}^{[1]})$ for all $u \in [W_1]$ and $v \in [H_1]$, then the output function of two-layer CNNs reads

$$f_{\mathrm{CNN}}(\boldsymbol{x},\boldsymbol{\theta}) := \sum_{\beta=1}^M \left\langle \boldsymbol{a}_\beta, \sigma\left( \boldsymbol{x}_\beta^{[1]} \right) \right\rangle = \sum_{\beta=1}^M \sum_{u=1}^{W_L} \sum_{v=1}^{H_L} \boldsymbol{a}_{u,v,\beta} \cdot \sigma\left( \boldsymbol{x}_{u,v,\beta}^{[L]} \right),$$

except that in this case, the outmost inner product in the above equation is with dimension $W_1 H_1$, i.e., $\langle \cdot, \cdot \rangle : \mathbb{R}^{W_1 H_1} \times \mathbb{R}^{W_1 H_1} \to \mathbb{R}$. Hence, the number of channels $M$ in $\boldsymbol{x}^{[1]}(i)$ can be heuristically understood as the 'width' of the hidden layer in the case of two-layer NNs.

## 4 THEORY: CONDENSATION OF THE CONVOLUTIONAL KERNEL

In this section, we demonstrate that when subject to the small initialization scheme (Assumption 3) and under the finite spectral gap condition (Assumption 4), kernels within the same layer condense toward a specific direction within a period of time $T_{\mathrm{eff}}$. We consider the dataset with one input channel, i.e. $C_0 = 1$, and omit the third index in $\boldsymbol{W}$ in the following discussion. Multi-channel analysis is similar and is shown in the Appendix. D. We begin this part by some technical conditions (Zhou et al., 2022, Definition 1) on the activation function $\sigma(\cdot)$.

**Definition 1** (Multiplicity $r$). $\sigma(\cdot) : \mathbb{R} \to \mathbb{R}$ *has multiplicity $r$ if there exists an integer $r \geq 1$, such that for all $0 \leq s \leq r - 1$, the $s$-th order derivative satisfies $\sigma^{(s)}(0) = 0$, and $\sigma^{(r)}(0) \neq 0$.*

**Assumption 1** (Multiplicity 1). *The activation function $\sigma \in \mathcal{C}^2(\mathbb{R})$, and there exists a universal constant $C_L > 0$, such that its first and second derivatives satisfy*

$$\left\| \sigma^{(1)}(\cdot) \right\|_\infty \leq C_L, \quad \left\| \sigma^{(2)}(\cdot) \right\|_\infty \leq C_L. \tag{6}$$

*Moreover,*

$$\sigma(0) = 0, \quad \sigma^{(1)}(0) = 1. \tag{7}$$

**Remark 1.** *We remark that $\sigma$ has multiplicity 1. $\sigma^{(1)}(0) = 1$ can be replaced by $\sigma^{(1)}(0) \neq 0$, and we set $\sigma^{(1)}(0) = 1$ for simplicity, which can be easily satisfied by replacing the original activation $\sigma(\cdot)$ with $\frac{\sigma(\cdot)}{\sigma^{(1)}(0)}$.*

**Assumption 2.** *The training inputs $\{\boldsymbol{x}_i\}_{i=1}^n$ and labels $\{y_i\}_{i=1}^n$ satisfy that there exists a universal constant $c > 0$, such that given any $i \in [n]$, then for each $u \in [W_0]$, $v \in [H_0]$ and $\alpha \in [C_0]$, the following holds*

$$\frac{1}{c} \leq |\boldsymbol{x}_{u,v,\alpha}(i)|, \quad |y_i| \leq c.$$

We assume further that

**Assumption 3.** *The following limit exists*

$$\gamma := \lim_{M \to \infty} -\frac{\log \varepsilon^2}{\log M}. \tag{8}$$

We initialize the parameters following
$$\boldsymbol{W}_{p,q,\beta}^0 \sim \mathcal{N}(0, \varepsilon^2), \quad \boldsymbol{b}_\beta^0 \sim \mathcal{N}(0, \varepsilon^2), \quad \boldsymbol{a}_{u,v,\beta}^0 \sim \mathcal{N}(0, \varepsilon^2),$$
and as we denote

$$
\begin{aligned}
\boldsymbol{\theta}_\beta := \Big( &\boldsymbol{W}_{0,0,1,\beta}, \boldsymbol{W}_{0,1,1,\beta}, \cdots, \boldsymbol{W}_{0,m-1,1,\beta}; \boldsymbol{W}_{1,0,1,\beta}, \cdots, \boldsymbol{W}_{1,m-1,1,\beta}; \cdots\cdots \boldsymbol{W}_{m-1,m-1,1,\beta}; \\
&\boldsymbol{W}_{0,0,2,\beta}, \boldsymbol{W}_{0,1,2,\beta}, \cdots, \boldsymbol{W}_{0,m-1,2,\beta}; \boldsymbol{W}_{1,0,2,\beta}, \cdots, \boldsymbol{W}_{1,m-1,2,\beta}; \cdots\cdots \boldsymbol{W}_{m-1,m-1,2,\beta}; \\
&\cdots\cdots\cdots\cdots\cdots\cdots\cdots\cdots\cdots\cdots\cdots\cdots\cdots\cdots\cdots\cdots\cdots\cdots\cdots\cdots\cdots \\
&\boldsymbol{W}_{0,0,d_0,\beta}, \boldsymbol{W}_{0,1,d_0,\beta}, \cdots, \boldsymbol{W}_{0,m-1,d_0,\beta}; \cdots, \boldsymbol{W}_{1,m-1,d_0,\beta}; \cdots\cdots \boldsymbol{W}_{m-1,m-1,d_0,\beta}; \boldsymbol{b}_\beta; \\
&\boldsymbol{a}_{1,1,\beta}, \boldsymbol{a}_{1,2,\beta}, \cdots, \boldsymbol{a}_{1,H_1,\beta}; \boldsymbol{a}_{2,1,\beta}, \cdots, \boldsymbol{a}_{2,H_1,\beta}; \cdots\cdots \boldsymbol{a}_{W_1,H_1,\beta} \Big)^\mathsf{T},
\end{aligned}
$$

and

$$\boldsymbol{\theta}_{\boldsymbol{W},\beta} := \left( \boldsymbol{W}_{0,0,\beta}, \boldsymbol{W}_{0,1,\beta}, \cdots, \boldsymbol{W}_{0,m-1,\beta}; \boldsymbol{W}_{1,0,\beta}, \cdots, \boldsymbol{W}_{1,m-1,\beta}; \cdots\cdots \boldsymbol{W}_{m-1,m-1,\beta}; \boldsymbol{b}_\beta \right)^\mathsf{T},$$

$$\boldsymbol{\theta}_{\boldsymbol{a},\beta} := \left( \boldsymbol{a}_{1,1,\beta}, \boldsymbol{a}_{1,2,\beta}, \cdots, \boldsymbol{a}_{1,H_1,\beta}; \boldsymbol{a}_{2,1,\beta}, \cdots, \boldsymbol{a}_{2,H_1,\beta}; \cdots\cdots \boldsymbol{a}_{W_1,H_1,\beta} \right)^\mathsf{T},$$

then $\boldsymbol{\theta}_\beta = \left( \boldsymbol{\theta}_{\boldsymbol{W},\beta}^\mathsf{T}, \boldsymbol{\theta}_{\boldsymbol{a},\beta}^\mathsf{T} \right)^\mathsf{T}$. We are able to identify the vectorized parameters $\boldsymbol{\theta}_\beta$ as variables of order 1 by setting $\boldsymbol{\theta}_\beta = \varepsilon \bar{\boldsymbol{\theta}}_\beta$, with the following initialization

$$\overline{\boldsymbol{W}}_{p,q,\beta}^0 \sim \mathcal{N}(0,1), \quad \bar{\boldsymbol{b}}_\beta^0 \sim \mathcal{N}(0,1), \quad \bar{\boldsymbol{a}}_{u,v,\beta}^0 \sim \mathcal{N}(0,1).$$

In the following discussion throughout this paper, we always refer to the above rescaled dynamics and drop all the "bar"s of $\bar{\boldsymbol{\theta}}$ for notational simplicity. For all $i \in [n]$, as we denote that $e_i := e_i(\boldsymbol{\theta}) := f(\boldsymbol{x}_i, \boldsymbol{\theta}) - y_i$, and under the condition $\gamma > 1$, $\varepsilon < \frac{1}{\sqrt{M}}$, hence as $\left| \sum_{\beta=1}^M \varepsilon^2 \left\langle \boldsymbol{a}_\beta, \sigma\left( \boldsymbol{x}_\beta^{[1]}(i) \right) \right\rangle \right| \leq M \log M \varepsilon^2 \ll 1$, the relation $e_i = \sum_{\beta=1}^M \varepsilon^2 \left\langle \boldsymbol{a}_\beta, \sigma\left( \boldsymbol{x}_\beta^{[1]}(i) \right) \right\rangle - y_i \approx -y_i$ holds. By means of perturbation expansion with respect to $\varepsilon$ with multiplicity $r = 1$, we obtain that

$$\frac{\mathrm{d} \boldsymbol{W}_{p,q,\beta}}{\mathrm{d}t} \approx \frac{1}{n} \sum_{i=1}^n y_i \cdot \left( \sum_{u=1}^{W_1} \sum_{v=1}^{H_1} \boldsymbol{a}_{u,v,\beta} \cdot \boldsymbol{x}_{u+p,v+q}(i) \right) = \sum_{u=1}^{W_1} \sum_{v=1}^{H_1} \boldsymbol{a}_{u,v,\beta} \cdot \boldsymbol{z}_{u+p,v+q}$$

$$\frac{\mathrm{d} \boldsymbol{b}_\beta}{\mathrm{d}t} \approx \frac{1}{n} \sum_{i=1}^n y_i \cdot \sum_{u=1}^{W_1} \sum_{v=1}^{H_1} \boldsymbol{a}_{u,v,\beta} = \sum_{u=1}^{W_1} \sum_{v=1}^{H_1} \boldsymbol{a}_{u,v,\beta} \cdot z,$$

$$\frac{\mathrm{d} \boldsymbol{a}_{u,v,\beta}}{\mathrm{d}t} \approx \frac{1}{n} \sum_{i=1}^n y_i \cdot \boldsymbol{x}_{u,v,\beta}^{[1]}(i) = \left( \sum_{p=0}^{m-1} \sum_{q=0}^{m-1} \boldsymbol{z}_{u+p,v+q} \cdot \boldsymbol{W}_{p,q,\beta} \right) + \boldsymbol{b}_\beta \cdot z.$$

as we set $\boldsymbol{z}_{u+p,v+q} := \frac{1}{n} \sum_{i=1}^{n} y_i \boldsymbol{x}_{u+p,v+q}(i)$ and $z := \frac{1}{n} \sum_{i=1}^{n} y_i$. To sum up, we approximate the initial GD dynamics by the following linear dynamics

$$\frac{\mathrm{d}\boldsymbol{\theta}_\beta}{\mathrm{d}t} = \boldsymbol{A}\boldsymbol{\theta}_\beta, \tag{9}$$

with

$$\boldsymbol{A} := \begin{bmatrix} \boldsymbol{0}_{(C_0 m^2 + 1) \times (C_0 m^2 + 1)} & \boldsymbol{Z}^\mathsf{T} \\ \boldsymbol{Z} & \boldsymbol{0}_{W_1 H_1 \times W_1 H_1} \end{bmatrix}, \tag{10}$$

and $\boldsymbol{Z}$ is detailed described by multi channel (70) and single channel (35) in appendix, whose entries consist of $\boldsymbol{z}_{u+p,v+q}$ and $z$. By performing singular value decomposition (SVD) on $\boldsymbol{Z}$,

$$\boldsymbol{Z} = \boldsymbol{U}\Lambda\boldsymbol{V}^\mathsf{T}, \tag{11}$$

where

$$\boldsymbol{U} = [\boldsymbol{u}_1, \boldsymbol{u}_2, \cdots, \boldsymbol{u}_{W_1 H_1}], \quad \boldsymbol{V} = [\boldsymbol{v}_1, \boldsymbol{v}_2, \cdots, \boldsymbol{v}_{m^2+1}],$$

and as we denote $r := \mathrm{rank}(\boldsymbol{Z})$, naturally, $r \leq \min\{W_1 H_1, m^2 + 1\}$, we have $r$ singular values,

$$\lambda_1 \geq \lambda_2 \geq \cdots \geq \lambda_r > 0.$$

We remark that for two-layer NNs, as $\boldsymbol{Z}$ "degenerates" to vector $\boldsymbol{z} := \frac{1}{n} \sum_{i=1}^{n} y_i \boldsymbol{x}_i$ (Chen et al., 2023), hence the rank $r \leq \min\{W_1 H_1, 1\}$ is at most equals to 1, or even 0 if $\boldsymbol{z} = \boldsymbol{0}$. We proceed to impose a technical condition on $\boldsymbol{Z}$ to ensure that the kernels $\boldsymbol{\theta}_\beta$ condense toward the direction of the largest eigenvector $\boldsymbol{v}_1$.

**Assumption 4** (Spectral Gap of $\boldsymbol{Z}$). *The singular values $\{\lambda_k\}_{k=1}^{r}$ of $\boldsymbol{Z}$ satisfy that*

$$\lambda_1 > \lambda_2 \geq \cdots \geq \lambda_r > 0, \tag{12}$$

*and we denote the spectral gap between $\lambda_1$ and $\lambda_2$ by*

$$\Delta\lambda := \lambda_1 - \lambda_2.$$

We remark that in the case of two-layer NNs, the spectral gap is always finite as long as the vector $\boldsymbol{z} = \frac{1}{n} \sum_{i=1}^{n} y_i \boldsymbol{x}_i \neq \boldsymbol{0}$ is non-zero (Assumption 3 in (Chen et al., 2023).) We also remark that the experiments conducted in Fig. 2a serves to demonstrate the universality of finite spectral gap for some commonly used datasets such as CIFAR10.

In order to study the phenomenon of condensation, we concatenate the vectors $\{\boldsymbol{\theta}_{\boldsymbol{W},\beta}\}_{\beta=1}^{M}$ into $\boldsymbol{\theta}_{\boldsymbol{W}} := \mathrm{vec}\left(\{\boldsymbol{\theta}_{\boldsymbol{W},\beta}\}_{\beta=1}^{M}\right)$, and we denote further that

$${\color{red}\boldsymbol{\theta}_{\boldsymbol{W},\boldsymbol{v}_1} := (\langle \boldsymbol{\theta}_{\boldsymbol{W},1}, \boldsymbol{v}_1 \rangle, \langle \boldsymbol{\theta}_{\boldsymbol{W},2}, \boldsymbol{v}_1 \rangle, \cdots \langle \boldsymbol{\theta}_{\boldsymbol{W},M}, \boldsymbol{v}_1 \rangle)^\mathsf{T},}$$

where $\boldsymbol{v}_1$ is the eigenvector of the largest eigenvalue of $\boldsymbol{Z}^\mathsf{T}\boldsymbol{Z}$, or the first column vector of $\boldsymbol{V}$. In appendix Appendix C.5, we prove that for any $\eta_0 > \frac{\gamma-1}{100} > 0$, there exists $T_{\mathrm{eff}}$ satisfying

$$T_{\mathrm{eff}} > \frac{1}{\lambda_1}\left[\log\left(\frac{1}{4}\right) + \left(\frac{\gamma-1}{4} - \eta_0\right)\log(M)\right], \tag{13}$$

We observe that $T_{\mathrm{eff}}$ is of order at least $\log(M)$, and we hereby present a heuristic explanation. For the linear dynamics (9), as its solution reads $\boldsymbol{\theta}_\beta(t) = \exp(t\boldsymbol{A})\boldsymbol{\theta}_\beta(0)$, consequently for $\boldsymbol{\theta}_{\boldsymbol{W},\beta}$, its solution approximately takes the form $\boldsymbol{\theta}_{\boldsymbol{W},\beta}(t) \approx \sum_{k=1}^{r} \exp(t\lambda_k) \langle \boldsymbol{\theta}_{\boldsymbol{W},\beta}(0), \boldsymbol{v}_k \rangle \boldsymbol{v}_k$. Then, under the finite spectral gap condition, the direction of $\boldsymbol{v}_1$ is the direction of attraction for $\boldsymbol{\theta}_{\boldsymbol{W},\beta}$ since it has the largest exponential growth rate. Moreover, in order for the condensation phenomenon to be observed, it is required for $\boldsymbol{\theta}_\beta$ to grow from order one at time $t = 0$, i.e., $\boldsymbol{\theta}_\beta(0) \approx M^0$, to the order of $\delta$ i.e., $\boldsymbol{\theta}_\beta(t) \approx M^\delta$, for some $\delta > 0$, while still maintaining the asymptotic relation $e_i \approx -y_i$. Since $\gamma > 1$ based on Assumption 3, then as we choose $\delta = \frac{\gamma-1}{4}$, we guarantee the existence of $\delta$. Consequently, as $\boldsymbol{v}_1$ is the direction of dominance, $\boldsymbol{\theta}_{\boldsymbol{W},\beta}(T) \approx \exp(T\lambda_1)\boldsymbol{\theta}_{\boldsymbol{W},\beta}(0)$, i.e., $M^\delta \approx \exp(T\lambda_1)M^0$, it takes time at least of order $T \approx \frac{\delta}{\lambda_1}\log M$ for the condensation phenomenon to be observed.

**Theorem 1.** *Given any $\delta \in (0,1)$, under Assumption 1, Assumption 2, Assumption 3 and Assumption 4, if $\gamma > 1$, then with probability at least $1 - \delta$ over the choice of $\boldsymbol{\theta}^0$, we have*

$$\lim_{M \to +\infty} \sup_{t \in [0, T_{\text{eff}}]} \frac{\|\boldsymbol{\theta}_{\boldsymbol{W}}(t) - \boldsymbol{\theta}_{\boldsymbol{W}}(0)\|_2}{\|\boldsymbol{\theta}_{\boldsymbol{W}}(0)\|_2} = +\infty, \tag{14}$$

*and*

$$\lim_{M \to +\infty} \sup_{t \in [0, T_{\text{eff}}]} \frac{\|\boldsymbol{\theta}_{\boldsymbol{W}, \boldsymbol{v}_1}(t)\|_2}{\|\boldsymbol{\theta}_{\boldsymbol{W}}(t)\|_2} = 1. \tag{15}$$

The above theorem demonstrates in the settings of overparametrization ($M \to \infty$), within a finite period $T_{\text{eff}}$, the relative change of kernel weight tends to infinity, while the kernel weight concentrates on a specific direction $\boldsymbol{v}_1$ determined by the training samples.

Fig. 2a presents an illustrative example of the eigenvalues of $\boldsymbol{Z}^\intercal \boldsymbol{Z}$ for the CIFAR10 dataset. Notably, a substantial spectral gap is observed, thereby satisfying Assumption 4. Subsequently, we delve into the study of the eigenvector corresponding to the maximum eigenvalue. Given that the input sample comprises 3 channels, we decompose the eigenvector of the largest eigenvalue into three corresponding parts. Our investigation reveals that the inner product of each part with $\mathbb{1}$ is approximately 1, with values of $0.9891 \pm 0.0349$, $0.9982 \pm 0.0009$, and $0.9992 \pm 0.0003$, respectively, computed from 50 independent trials, where each trial involves the random selection of 500 images. Based on these findings, we predict that the convolutional kernels have a propensity to condense towards the direction of $\mathbb{1}$, given that $\mathbb{1}$ is the eigenvector of the largest eigenvalue. As validated in Fig. 2, our prediction holds true. Thus, we can confidently conclude that this example effectively demonstrates how the theoretical framework guides the experimental investigations. The spectral gap for MNIST is displayed in Fig. 12.

Through our careful analysis, we discovered that the primary difference between condensation in fully-connected neural networks and CNNs at the initial stage is that in fully-connected neural networks, condensation occurs among different neurons within a given layer (Zhou et al., 2022), whereas in CNNs, condensation arises across different convolutional kernels within each convolution layer. This difference in condensation is mainly caused by the structure of the local receptive field and weight-sharing mechanism in CNNs.

## 5 EXPERIMENTS: CONDENSATION OF THE CONVOLUTIONAL KERNELS

In the following section, we present an extensive empirical analysis to enhance the understanding of the condensation phenomenon among convolutional kernels. Our investigation particularly focuses on several key aspects, including the selection of activation functions, loss functions, and optimizers, as well as the utilization of different image datasets. The primary objectives of our experiments are twofold. Firstly, they serve as a means to substantiate the theoretical framework we have established, which primarily focus on two-layer convolutional neural networks trained with gradient descent. Secondly, our experimental design extends beyond the confines of our theoretical analysis. Remarkably, we observe that CNNs with three convolution layers also exhibit similar kernel condensation behavior when subjected to alternative first-order optimization methods such as ADAM. These empirical findings provide valuable insights into the universal occurrence of of kernel condensation phenomena across diverse training settings and optimization methods.

### 5.1 EXPERIMENTAL SETUP

For the CIFAR10 dataset: 500 samples are randomly selected from CIFAR10 dataset for training. The used CNN has $H$ convolution layers, followed by an output layer with $d$ neurons in Fig. 2b and an extra fully-connected hidden layer with 1024 neurons between the convolution layers and the output layer in Fig. 3 and Fig. 4. Each convolution layer has 32 channels. The output dimension $d = 10$ or $1$ is used for the classification problem or for the regression problem, respectively. The parameters of the convolution layer is initialized by the $\mathcal{N}(0, \sigma_1^2)$, and the parameters of the linear layer is $\mathcal{N}(0, \sigma_2^2)$. $\sigma_1$ is given by $\left(\frac{(c_{in} + c_{out}) * m^2}{2}\right)^{-\gamma}$ where $c_{in}$ and $c_{out}$ are the number of in channels and out channels respectively, $\sigma_2$ is given empirically by $0.0001$. The training method is GD or Adam with full batch and learning rate $lr$. The training loss of each experiment is shown in Fig. 13 in Appendix.

## 5.2 CIFAR10 EXAMPLES

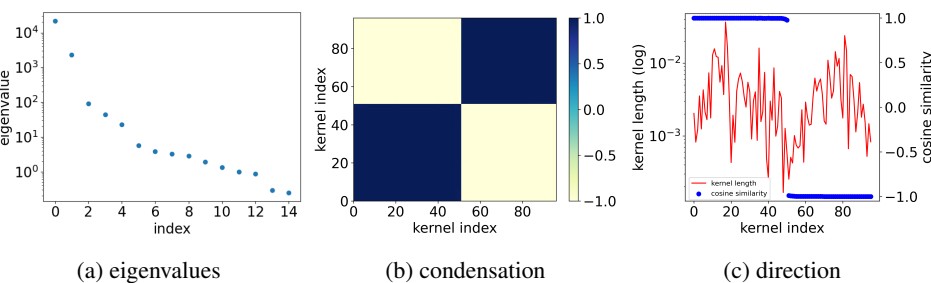

(a) eigenvalues   (b) condensation   (c) direction

Figure 2: Left: The largest 15 eigenvalues of $Z^{\mathsf{T}}Z$ of CIFAR10 dataset. A clear spectral gap ($\Delta\lambda := \lambda_1 - \lambda_2$) could be observed, which satisfies Assumption 4 and leads to condensation in Theorem 1. Middle: Condensation of two-layer CNNs. The activation functions are $\tanh(\mathrm{x})$. The numbers of step selected is epoch = 4500 with accuracy less than 20%. The color indicates $D(\boldsymbol{u}, \boldsymbol{v})$ of two different kernels, whose indexes are indicated by the abscissa and the ordinate, respectively. If kernels are in the same beige block, $D(\boldsymbol{u}, \boldsymbol{v}) \sim 1$ (navy-blue block, $D(\boldsymbol{u}, \boldsymbol{v}) \sim -1$), their input weights have the same (opposite) direction. Right: Left ordinate (red): the amplitude of each kernel; Right ordinate (blue): cosine similarity between kernel weight and $\mathbb{1}$. The training data is CIFAR10 dataset. ~~ReLU is used for linear layer,~~ MSE for loss function and full batch GD for optimizer. $m = 3$, $lr = 5 \times 10^{-6}$, and $\gamma = 4$.

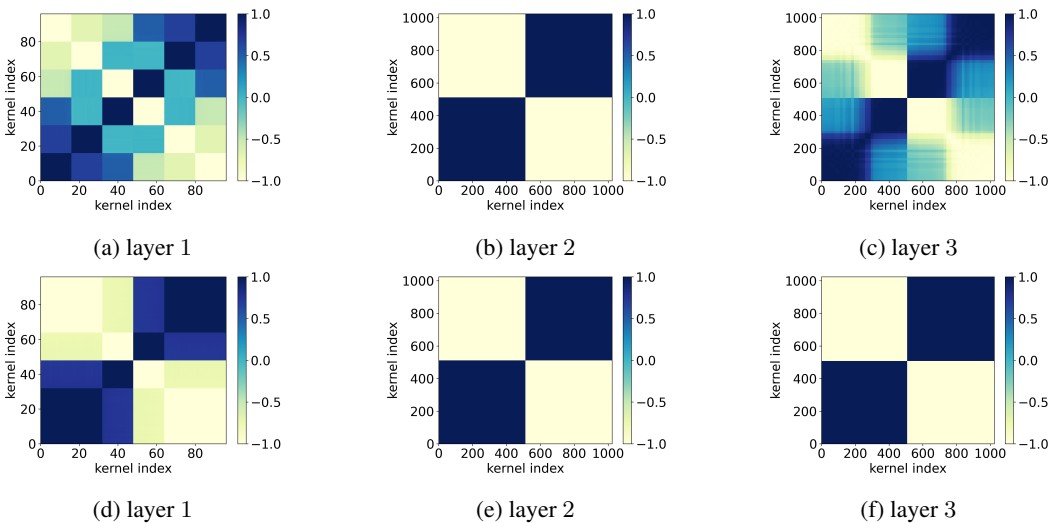

(a) layer 1   (b) layer 2   (c) layer 3

(d) layer 1   (e) layer 2   (f) layer 3

Figure 3: Condensation of three-convolution-layer CNNs. The activation functions are $\tanh(\mathrm{x})$. The numbers of steps selected in (a)-(c) are the final stage after training, while the numbers of steps selected in (d)-(f) are epoch = 300 with accuracy less than 20%. The NN is only trained once. The color indicates $D(\boldsymbol{u}, \boldsymbol{v})$ of two different kernels, whose indexes are indicated by the abscissa and the ordinate, respectively. The training data is CIFAR10 dataset. ReLU is used for linear layer, softmax for output layer, cross-entropy for loss function and full batch Adam for optimizer. $m = 5$, $lr = 2 \times 10^{-6}$, and $\gamma = 2$.

We first conduct the experiments under the theoretical setting, i.e., a one-layer convolutional neural network with the gradient descent (GD) method and MSE loss (softmax is also attached with the output layer), and verify the theoretical results. Fig. 2 illustrates that with GD, the kernel weights of a two-layer CNN undergo condensation at two opposite directions during training, which is consistent with our theory. Moreover, we observe that the direction of the condensation is $\boldsymbol{v} = \mathbb{1}$. Fig. 2(b) and (c) show that the cosine similarity between each kernel weight and $\mathbb{1}$ is almost equal to 1 or -1. Besides, three-convolution-layer mse examples with and without softmax are also shown in Fig. 6 and Fig. 7 in appendix.

Then, we try to further understand the condensation of CNN through more experiments. Experiments show that when initialized with small weights, the convolutional kernels of a $\tanh(x)$ CNN undergo condensation during the training process. As shown in Fig. 3(a)-(c), we train a CNN with three convolution layers by cross-entropy loss until the training accuracy reaches $100\%$ (the accuracy during training is shown in Fig. 5 in appendix). In each layer, we compute the cosine similarity between each pair of kernels. This reveals a clear condensation phenomenon after training.

Understanding the mechanism of condensation phenomenon is challenging, no matter experimentally or theoretically. To this end, we still focus on the initial training stage. We then study the initial condensation in CNNs by more experiments.

The initial stage (accuracy less than $20\%$) of Fig. 3(a)-(c) is shown in Fig. 3(d)-(f). For each layer, all kernels are nearly condensed into two opposite directions in Fig. 3(d)-(f).

We further examine the different activation functions. For illustration, we consider two-layer CNNs with activations $\mathrm{ReLU}(x)$, $\mathrm{Sigmoid}(x)$, or $\tanh(x)$. As shown in Fig. 4, we can still see a very clear condensation phenomenon. Note that, as the learning rate is rather small to see the detailed training process, the epoch selected for studying the initial stage may appear large.

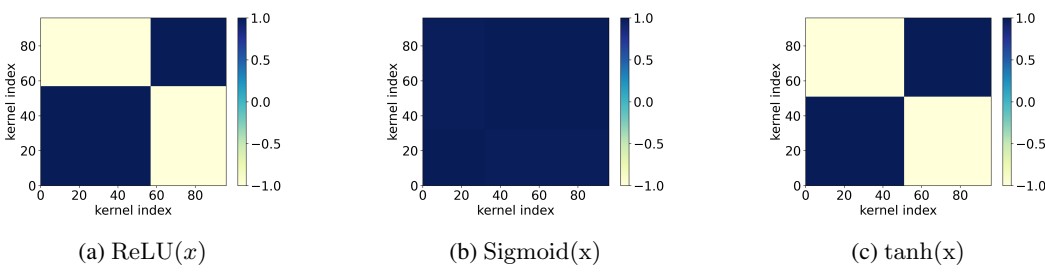

(a) $\mathrm{ReLU}(x)$        (b) $\mathrm{Sigmoid(x)}$        (c) $\tanh(\mathrm{x})$

Figure 4: Condensation of two-layer CNNs. The activation functions are indicated by the sub-captions. The numbers of steps selected in the sub-pictures are epoch$= 1000$, epoch$= 5000$ and epoch$= 300$, respectively. The color indicates $D(\boldsymbol{u}, \boldsymbol{v})$ of two different kernels, whose indexes are indicated by the abscissa and the ordinate, respectively. The training data is CIFAR10 dataset. ReLU is used for linear layer, softmax for output layer, cross-entropy for loss function and full batch Adam for optimizer. $m = 5$, $lr = 5 \times 10^{-7}$, and $\gamma = 2$.

Similar results of MNIST dataset are also shown in Figs. 8 for CNNs with different activations, Fig. 9 for three-convolution-layer $\tanh$ CNN in appendix. Also, two-layer CNNs with 32 and 320 kernels trained by GD on MNIST are shown in Fig.10 and Fig.11, respectively, in appendix.

Taken together, our empirical analysis provides compelling evidence that when subjected to small initialization, kernel weights within the same layer of a three-convolution-layer CNN tend to cluster together during the training process. These consistent findings across various activation functions and optimization methods confirm the existence of the condensation phenomenon in simple CNNs.

## 6   Conclusion

In this work, our theoretical analysis demonstrate that under GD and small initialization, kernels of a two-layer CNN condense towards a specific direction determined by the training samples within a given time period. These theoretical findings have been substantiated through extensive empirical validations.

Furthermore, our experimental results exceed the theoretical findings as they confirm the condensation phenomenon in simple CNNs across various activation functions and optimization methods. These empirical findings have opened up new avenues for further exploration and analysis of condensation in more general CNN architectures.

In summary, we contribute to a deeper understanding of the condensation phenomenon in CNNs by presenting a preliminary theory for two-layer CNNs, and validate the possibility for future exploration of condensation in multi-layer CNNs through systematic empirical investigations.

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
