# OpenReview forum: "Understanding the Initial Condensation of Convolutional Neural Networks"
_ICLR.cc/2024/Conference — Submitted to ICLR 2024_

### Official Review · Reviewer_mFgA · 2023-10-16

**Soundness:** 2 fair
**Presentation:** 1 poor
**Contribution:** 2 fair
**Rating:** 3
**Confidence:** 2

**Summary:**

This paper analyzes the phenomenon called initial condensation in simple CNN models. Initial condensation refers to the occurrence of weight grouping in the early stages of neural network (NN) training, which has been discussed in previous literature mainly with respect to fully connected NNs. A two-layer CNN is theoretically analyzed and it is shown that initial condensation occurs in one CNN layer with respect to convolutional kernels.

**Strengths:**

- Initial condensation in CNN models is analyzed theoretically, although the assumed model is restricted to a simple two-layer one.

**Weaknesses:**

- It is difficult to follow the technical details of the paper for the following reasons:
    - $W_{p,q,\alpha,\beta}^{[l]}$ is not well defined for $p=-\infty, \ldots, \infty$ and $q=-\infty, \ldots, \infty$, although it is required to define $x_{u,v,\beta}^{[l]}$.
    - I could not understand the meaning of $\frac{dW_{p,q,\beta}}{dt}$. What is $t$?
    - The description of the paper does not follow the notations defined in Section 3.1. For example, the matrix notation introduced in Section 3.1 is not used in the main technical discussion in Section 4. Also, the operator norm is defined in Section 3.1, but does not appear in the manuscript.

**Questions:**

- Why are only 500 samples used in the CIFAR10 experiment?

- What is the test accuracy of the final model in Figure 3? I wonder if the final model overfits the training set.

- It is better to use \citet{} for textual citation.

- It is better to avoid using multiple $:=$ in one line.

- In page 4: .. -> .

---

> ### Author Response · Authors · 2023-11-18
> **Response to Reviewer mFgA**
>
> $\textbf{Point 1:}$
>
> It is difficult to follow the technical details of the paper for the following reasons:
>
> 1. $\boldsymbol{W}_{p,q,\alpha,\beta}^{[l]}$is not well defined for $p=\infty,\infty,\cdots,\infty$ and $q=\infty,\infty,\cdots,\infty$, although it is required to define.
>
> 2. I could not understand the meaning of $\frac{d\boldsymbol{W}_{p,q,\beta}}{dt}$. What is $t$?
>
> 3. The description of the paper does not follow the notations defined in Section 3.1. For example, the matrix notation introduced in Section 3.1 is not used in the main technical discussion in Section 4. Also, the operator norm is defined in Section 3.1, but does not appear in the manuscript.
>
> $\textbf{Reply 1:}$
>
> 1. We disagree that it is necessary to define $W_{p,q,\alpha,\beta}^{[l]}$ for $p=\infty,\infty,\cdots,\infty$ and $q=\infty,\infty,\cdots,\infty$. This is a notation for the weight of convolution kernels, where $p,q$ represent the weight and height of a convolution kernel. And these values will not reach $\infty$ in the real convolutional networks.
>
> 2. We apologize for any confusion regarding the term $\(\frac{dW_{p,q,\beta}}{dt}\)$. In our theoretical framework, we base our analysis on the concept of gradient flow, which is the continuous form of gradient descent. Here, \(t\) represents time in the continuous sense. The gradient flow is formulated as $\(-\nabla_{\theta}f(x;\boldsymbol{\theta}) $$ = \lim\limits_{\eta\rightarrow 0}\frac{\theta_{k+1}-\theta_{k}}{\eta}=\frac{d\theta}{dt}\)$. This allows us to express $\(\frac{d\boldsymbol{W}_{p,q,\beta}}{dt}\)$ in terms of $\(t\)$, with the understanding that in the context of training neural networks with a sufficiently small learning rate, it can be approximated as  $\(\eta \times \text{epochs}\)$.
>
> 3. The notation such as $\(\Vert \cdot \Vert_2\)$ and $\(\Vert \cdot \Vert_{2\to 2}\)$ is indeed critical to the theoretical derivations in the appendix. While they may not appear prominently in the main technical discussion in Section 4, their inclusion is essential for maintaining the coherence and completeness of our paper. We have reviewed our manuscript to ensure that the notations introduced in Section 3.1 are consistently applied and clearly articulated throughout the paper.
>
> $\textbf{Point 2:}$
>
> Why are only 500 samples used in the CIFAR10 experiment?
>
> $\textbf{Reply 2:}$
>
> Here are two key reasons. One, our theory demonstrates the condensation phenomenon at the initial stage is independent of training set size. Two, 500 samples can save a lot of computational cost for performing a large set of experiments.
>
> $\textbf{Point 3:}$
>
> What is the test accuracy of the final model in Figure 3? I wonder if the final model overfits the training set.
>
> $\textbf{Reply 3:}$
>
> Since only $500$ samples from CIFAR10 were employed to train the model, the test error can not give any useful information. This experiment is to show the condensation of final learning stage rather than the model's generalization.
>
> For your reference, we have found that many pretrained large neural networks also exhibit clear condensation, which are easy to be verified.
>
> $\textbf{Point 4:}$
>
> It is better to use citet\{\} for textual citation.
>
> $\textbf{Reply 4:}$
>
> Thanks for your advice, and we have changed the textual citation with citet\{\}.
>
> $\textbf{Point 5:}$
>
> It is better to avoid using multiple $:=$ in one line.
>
> $\textbf{Reply 5:}$
>
> Thanks for your advice, and we have modified the mathematical expression in Page 6.

---

> > ### Comment · Reviewer_mFgA · 2023-11-22
> > **Thank you for your responses**
> >
> > Thank you for your responses.
> >
> > I still do not understand why the definition of $W_{p,q,\alpha,\beta}^{[l]}$ is not necessary for $p=-\infty, \ldots, \infty$ and $q=-\infty, \ldots, \infty$, even though they are summed over $p=-\infty, \ldots, \infty$ and $q=-\infty, \ldots, \infty$.
> >
> > I am still confused about matrix notations.
> > For example, in Section 3.1 it is defined to use $A_{i,:}$ to denote the $i$-th row of a matrix $A$.
> > In Assumption 3, however, both $W_{0,0,1,\beta}$ and $W_{0,0,\beta}$ appear. What do these expressions mean?
> >
> > If notations in Section 3.1 are only used in the appendix, it is better to define them in the appendix.
> >
> > > we have modified the mathematical expression in Page 6.
> >
> > I have checked the current version of your manuscript and find no update on page 6.
> >
> > I would like to keep my score.

---

> ### Author Response · Authors · 2023-11-23
> **Response to Reviewer mFgA**
>
> We are sorry that we have forgotten to upload the revised manuscript, and we have uploaded the newest version with revision including the current reply.
>
> $\textbf{Point 1:}$
> I still do not understand why the definition of $\boldsymbol{W}_{p,q,\alpha,\beta}^{[l]}$
> is not necessary for $p=\infty,\infty,\cdots,\infty$ and $q=\infty,\infty,\cdots,\infty$ , even though they are summed over $p=\infty,\infty,\cdots,\infty$ and $q=\infty,\infty,\cdots,\infty$.
>
> $\textbf{Reply 1:}$
>
> As we define at the last of page 3 that
>
> $$
>     \chi (p, q) =
> \begin{array} {l}1,\\ \\ \text{for~~} 0 \leqslant p, q \leqslant m-1
>     %%%%%%%%%%%%%%%%%%%%%%%%%%%%
>     \\\\ 0, \\ \\  \text{otherwise,} \end{array}
> $$
>
> we have the term in summation outside $0$ and  $m-1$ is zero for both $p$ and $q$. And the summation over $p=\infty,\infty,\cdots,\infty$ and $q=\infty,\infty,\cdots,\infty$ is just for a convenient writing. Thus the definition of $\boldsymbol{W}_{p,q,\alpha,\beta}^{[l]}$ is not necessary for $p=\infty,\infty,\cdots,\infty$ and $q=\infty,\infty,\cdots,\infty$.
>
> $\textbf{Point 2:}$
>
> I am still confused about matrix notations. For example, in Section 3.1 it is defined to use $A _{i,:}$ to denote the $i$-th row of a matrix $A$. In Assumption 3, however, both $\boldsymbol{W} _{0,0,1,\beta}$ and $\boldsymbol{W} _{0,0,\beta}$ appear. What do these expressions mean?
>
> If notations in Section 3.1 are only used in the appendix, it is better to define them in the appendix.
>
> $\textbf{Reply 2:}$
> Thanks for your advice and we have modified the Section 3.1, and we have also added some notations for high-dimensional tensors, for example, a four-dimensional tensor $\boldsymbol{W} _{i,j,k,l}$ which may help understanding.
>
> $\boldsymbol{W} _{0,0,1,\beta}$ is for multiple input channels and  $\boldsymbol{W} _{0,0,\beta}$ is for one input channel. In our revised manuscript, we clarify at the last of page 4 that "We consider the dataset with one input channel, i.e. $C _0 = 1$, and omit the third index in $\mathbf{W}$ in the following discussion. Multi-channel analysis is similar and is shown in the Appendix. D. " And we modify the notation $\boldsymbol{\theta} _{\beta}$ on page 5.
> Since the case of multiple input channel is only analyzed in appendix, we have remove this notation (along with those only used in appendix) to appendix and only keep $\boldsymbol{W} _{0,0,\beta}$ in main text.

---

### Official Review · Reviewer_FvGf · 2023-10-27

**Soundness:** 3 good
**Presentation:** 1 poor
**Contribution:** 1 poor
**Rating:** 3
**Confidence:** 3

**Summary:**

This manuscript presents a study of condensation in convolutional neural networks.
The main theorem states that under certain assumptions on the data, activation function and initial weights, the following two things hold:
 1. the final weights go arbitrarily far away from the starting point
 2. the final weights all point in the direction of the principal eigenvector of some data-dependent matrix.

The experiments confirm the theoretical results, even in settings where assumptions are broken.

**Strengths:**

- (significance) Insights in the learning dynamics of neural networks typically help to guide model development and speed up learning.
 - (originality) The condensation problem has been studied for fully-connected networks, but this appears to be the first work on convolutional networks.

**Weaknesses:**

- (clarity) The paper is quite chaotic and therefore hard to read.
   Especially the frequent notation changes and notation that is used only in one place make the paper hard to read.
   E.g.&nbsp;something like $$\boldsymbol{\theta}_{\boldsymbol{W}, \boldsymbol{v}_1} := \operatorname{vec}\bigl(\\{\boldsymbol{\theta}\_{\boldsymbol{W},\beta} \cdot \boldsymbol{v}\_1\\}\_{\\beta=1}^M\bigr)$$
   would be much clearer than the current formula above equation&nbsp;(13).
 - (clarity) The variables $\eta_0$ and $T_\mathrm{eff}$ come out of nowhere and no intuition or explanation is provided about what these variables represent.
 - (clarity) The experiment section mentions that every CNN has an additional 2 fully-connected layers to produce outputs.
   As a result I do not understand how it is possible to do experiments with the theoretical setting which only considers convolutional networks.
 - (significance) I am unable to distill from the manuscript whether condensation is a good thing or a bad thing.
   Figure&nbsp;1 seems to suggest condensation enables learning smaller networks.
   However, the main theorem implies that only two possible directions of weights survives, which intuitively feels like a bad thing that would hinder expressivity.
   As a result, I also do not quite understand why having experiments where the assumptions are violated is supposed to be a selling point (unless it is a good thing).
   On the other hand, if it were a good thing, I am concerned about the caption of Figure&nbsp;2 and&nbsp;3 where it is stated that the network attains less than 20% accuracy.
 - (originality) It is not clear which parts of the analysis are taken from prior work and what new insights are necessary to make this work for convolutions.
   The use of the eigenvectors seems to be one of the most obvious differences
   but it would be good to highlight where exactly the differences are.
 - (quality) I am unable to properly assess the derivatiations and proofs because I understand too little of what is going on.
   However, I shortly skimmed over the proof of the main theorem and noticed a transition where the norm of sums becomes the sum of norms without any comments.
   Also, some non-obvious statements seem to be planted without proper explanation.
 - (quality) The theoretical results seems to build on an analysis of the dynamics of gradient descent.
   However, the experiments make use of adaptive optimisers like Adam, which should lead to significantly different dynamics than plain GD.

 ### Minor Comments

 - The hyperlinks in the paper seem to be broken.
   Reading the paper required more scrolling than I'm used to.
 - Assumption 4 seems to be more of a definition than an assumption.
 - I don't quite understand what the infinity norms are supposed to do in equation&nbsp;(6).

**Questions:**

1. Please, reduce the noise in the mathematical notation for the sake of readability.
    Note that there is more noise than the one example I provided (e.g. $\boldsymbol{x}_r$ and $\boldsymbol{w}_r$, $\boldsymbol{\theta}_\beta$, $\boldsymbol{U}$ and $\boldsymbol{V}$, ...)!
 2. Why is $\Big|\sum\_{\beta} \varepsilon \big\langle \boldsymbol{a}\_\beta, \sigma\bigl(\boldsymbol{x}\_\beta^{[1]}(i)\bigr)\big\rangle\Big| \leq M$ ?
 3. Where is the activation function in the time derivatives of the parameters?
    E.g. I would have expected the following for the derivative of the parameters in the last layer: $$\frac{\operatorname{d}\\!\boldsymbol{a}\_{u,v,\beta}}{\operatorname{d}\\!t} \approx \frac{1}{n} \sum_{i=1}^n y\_i \cdot \sigma\bigl(\boldsymbol{x}\_{u,v,\beta}^{[1]}(i)\bigr).$$
 4. Why is the supremum necessary in Theorem&nbsp;1?
    Is there a chance that condensation stops again before $T_\mathrm{eff}$?
 5. Do the assumptions correspond to the condensation regime from (Luo et al., 2021)?
    If not, what do the assumptions stand for?
 6. In which exact steps does this analysis differ from the analysis for fully-connected networks?
    It seems like the analyses have a lot in common.
 7. Is condensation a good or a bad thing?
 8. Why are the experiments conducted with networks that attain less than 20% accuracy?
 9. Does the theoretical setting in the experiment section also have 2 fully-connected layers at the end (as claimed in the experimental setup section)?
 10. Can the theory be directly applied to adaptive gradient methods, as used in the experiments?
 11. Does condensation also occur when the last layer is initialised with zeros?
 12. If the results apply when assumptions are violated, shouldn't it be possible to loosen the assumptions?

---

> ### Author Response · Authors · 2023-11-18
> **Response to Reviewer FvGf**
>
> $\textbf{Point 1:}$
> (clarity) The paper is $\dots$ considers convolutional networks
>
> $\textbf{Reply 1:}$
> First of all, we agree that a consistent and clear notation, such as using $\\theta_{W,v_1} := \operatorname{vec}({\theta_{W,\beta}\cdot v_1})\)$, would enhance the readability, particularly around equation (13). We are committed to revising the manuscript to ensure that the notation is more coherent and understandable throughout the text.
>
> Besides, for the abrupt introduction of variables $\(\eta_0\)$ and $\(T_{eff}\)$, we recognize the need for providing better context and explanation for these terms. In the revised manuscript, we have included a detailed description of $\(T_{eff}\)$ as the time period relevant to relation (13) and crucial to our asymptotic model's applicability. Similarly, $\(\eta_0\)$, which appears in appendix C.5, have been explained more thoroughly to enhance the reader's understanding.
>
> At last, the error in Section 5.1 about the CNN structure was unintentional. To clarify, only the experiment depicted in Figure 3 involves two fully connected layers. In contrast, the experiments in Figure 2 use a CNN with $H$ convolutional layers followed directly by an output layer with $d$ neurons, aligning with our theoretical framework. Additionally, the network used in Figure 4 only has a different optimizer than that in theory, which we have clarified in the revised manuscript.
>
> $\textbf{Point 2:}$
>
> (significance) I am unable to distill from the manuscript whether condensation is a good thing or a bad thing. ... On the other hand, if it were a good thing, I am concerned about the caption of Figure 2 and 3 where it is stated that the network attains less than 20\% accuracy.
>
> Question 7: Is condensation a good or a bad thing?
>
> Question 8: Why are the experiments conducted with networks that attain less than 20\% accuracy?
>
> $\textbf{Reply 2:}$
>
> As outlined in our "Introduction" section, condensation plays a crucial role in reducing the initial complexity of over-parameterized neural networks. This reduction in complexity is significant from the standpoint of complexity theory (referenced in Bartlett and Mendelson, 2002), as it contributes to our understanding of why such networks often exhibit good generalization performance despite being over-parameterized.
>
> The reviewer wonders how condensation on only two directions can be a good thing. Here is an explanation. This paper focuses on the initial stage. In this stage, condensation on two directions plays a role that resets the neural network to a simple state, whose expressivity is small. Then, due to the driven of non-zero loss, we find in the experiments that the network would condense on more and more directions in order to increase the expressivity. Such condensation with more and more directions can ensure the network to fit the target data points but with as low complexity as it can. Therefore, the initial condensation is an extremely important feature for neural network to well fit data in over-parameterized situation.
>
> Since we only study the initial stage as indicated by our paper title, it is then no surprise why the network attains less than 20\% accuracy.
>
> Our experiments show that the conclusion from the theory can also applies to the more general settings. This is a good thing because this indicates we can develop a similar theory for more general cases.
>
> $\textbf{Point 3:}$
>
> (originality) It is not clear which parts of the analysis are taken from prior work and what new insights are necessary to make this work for convolutions. The use of the eigenvectors seems to be one of the most obvious differences but it would be good to highlight where exactly the differences are.
>
> Question 6: In which exact steps does this analysis differ from the analysis for fully-connected networks? It seems like the analyses have a lot in common.
>
> $\textbf{Reply 3:}$
>
> The difference between FCNs and CNNs compels us to provide an additional commentary in the concluding section of Section 4  that ‘Through our careful analysis, we discovered that the primary difference between condensation in fully-connected neural networks and CNNs at the initial stage is that in fully-connected neural networks, condensation occurs among different neurons within a given layer (Zhou et al., 2022), whereas in CNNs, condensation arises across different convolution kernels within each convolution layer. This difference in condensation is mainly caused by the structure of the local receptive field and weight-sharing mechanism in CNNs’.
>
>  The exact step is that the weight-sharing mechanism in CNNs does indeed impact the structure of the matrix $A$ in their linearized dynamics, enabling more possible directions for the clustering effect of the convolution kernels. This multiplicity of directions may potentially exert impact on future training process of CNNs. Thus, it is essential for us to develop a condensation theory tailored to CNNs.

---

> ### Author Response · Authors · 2023-11-18
> **Response to Reviewer FvGf**
>
> $\textbf{Point 4:}$
>
> (originality) I am unable to properly assess the ... norms without any comments. Also, some non-obvious statements seem to be planted without proper explanation.
>
> $\textbf{Reply 4:}$
>
> We understand your concerns regarding the complexity of the derivations and proofs presented in our manuscript, particularly in the main theorem. It's crucial for our work to be accessible and comprehensible to our readers, including the detailed mathematical aspects.
>
> Regarding your specific observation about the transition from the norm of sums to the sum of norms, we realize that this step in our proof may not have been adequately explained, leading to potential misunderstandings. In our revised manuscript, we have ensured to clarify this transition more explicitly and provide the necessary mathematical justifications.
>
> Additionally, We have thoroughly reviewed these sections and augment them with more comprehensive explanations and justifications, to ensure that our mathematical reasoning is transparent and thoroughly grounded.
>
> $\textbf{Point 5:}$
>
> The theoretical results seems to build on an analysis of the dynamics of gradient descent. However, the experiments make use of adaptive optimisers like Adam, which should lead to significantly different dynamics than plain GD.
>
> Question 10: Can the theory be directly applied to adaptive gradient methods, as used in the experiments?
>
> $\textbf{Reply 5:}$
>
> Our theoretical analysis primarily focuses on gradient flow, which represents the continuous form of gradient descent. This approach allows for a rigorous and detailed understanding of the dynamics inherent in gradient descent algorithms.
>
> First of all, in Figure 2, the experimental parameters strictly follow the theoretical settings, which we did not state this clearly in the article at the beginning, but have now modified it. Besides, the objective of our experimental design, as detailed in Section 6, was to explore the boundaries of our theoretical predictions. We aimed to investigate whether the condensation phenomenon, which our theory predicts for simple CNNs under gradient flow, would still be observable under various activation functions and optimization methods, including adaptive ones. The results indeed confirm the presence of this phenomenon across these different settings, thus providing empirical support for the robustness and relevance of our theoretical findings, even in scenarios that extend beyond the specific conditions of our theoretical model.
>
> Therefore, while our theory is grounded in the context of gradient flow, the experimental outcomes suggest that the underlying principles may have broader applicability, including in settings involving adaptive gradient methods. We acknowledge, however, that a more detailed theoretical analysis tailored to these methods would be a valuable direction for future research.
>
> $\textbf{Point 6:}$
>
> Please, reduce the noise in the mathematical notation for the sake of readability. Note that there is more noise than the one example I provided (e.g. $x_r$ and $w_r$ $\theta_{\beta}$$UV$ , ...)!
>
> $\textbf{Reply 6:}$
>
> Thanks for your suggestion. We have changed the mathematical notation and make the manuscript more readable.
>
> $\textbf{Point 7:}$
>
> Why is $ \| {\sum_{\beta=1}^M \epsilon \left<a_{\beta}, \sigma\left(x_{\beta}^{[1]}(i)\right)\right> } \| \leq M.$
>
> $\textbf{Reply 7:}$
>
> I wonder if it referred to the inequality $\| {\sum_{\beta=1}^M \varepsilon ^2 \left <a_{\beta}, \sigma\left(x_{\beta}^{[1]}(i)\right)\right > } \| \leq M \varepsilon^2$ on page 5. We have that $\|{\sum_{\beta=1}^M\left<a_{\beta}, \sigma\left(x_{\beta}^{[1]}(i)\right)\right> } \| $ $\leq $ $ \sum_{\beta=1}^M $ $ \Vert a_{\beta} \Vert_2 $ $ \cdot \Vert \sigma \left(x_{\beta}^{[1]}(i)\right) \Vert_2 $ $\leq M H_1 W_1  \Vert a \Vert_{\infty} \Vert \sigma\left(x^{[1]}(i)\right) \Vert_{\infty}$ by triangle inequality and Schwartz inequality. And in our settings, since  $\boldsymbol{a}\sim\mathcal{N}(0,\varepsilon^2), \boldsymbol{W}\sim\mathcal{N}(0,\varepsilon^2),\boldsymbol{b} \sim\mathcal{N}(0,\varepsilon^2)$, then  by concentration of Gaussian variables, we obtain that with high probability, i.e., the probability of the  event is at least $1-\exp\left(-  M^{\alpha}\right)$ for some constant $\alpha>0$, the following holds: $\Vert a_{\beta} \Vert_{\infty} \leq $ $ \varepsilon \sqrt{\log M}\sim O(\varepsilon)$ and $\Vert \sigma \left(x_{\beta}^{[1]}(i)\right)\Vert_{\infty} $ $ = \Vert\sigma\left(Wx_{\beta}^{[0]}(i)+b\right)\Vert_{\infty} \leq \Vert Wx_{\beta}^{[0]}(i)+b \Vert_{\infty} \leq\varepsilon \sqrt{\log M}\sim O(\varepsilon)$. In the manuscript, we have already rescaled the parameters $a,W,\boldsymbol{b}$ to order $1$ and have taken the $\varepsilon$ of both elements outside the inner product to have $\varepsilon^2$ outside the $\left<\right>$. Also, compared with large $M$, the magnitude of $H_1$ and $W_1$ are both of order $1$. Thus the inequality on page 5 holds.

---

> ### Author Response · Authors · 2023-11-18
> **Response to Reviewer FvGf**
>
> $\textbf{Point 8: }$
>
> Where is the activation function in the time derivatives of the parameters? E.g. I would have expected the following for the derivative of the parameters in the last layer: $\frac{d a_{u,v,\beta}}{d t} \approx\frac{1}{n} \sum\limits_{i=1}^n y_i \sigma(x^{[1]}_{u,v,\beta}(i)) $
>
> $\textbf{Reply 8: }$
>
> We agree that  $\frac{d a_{u,v,\beta}}{d t}$ $ \approx $ $ \frac{1}{n} \sum \limits_{i=1}^n $ $ y_i \sigma $ $(x_{u,v,\beta}^{\[1\]}$ $(i)) $. In our settings, $a\sim\mathcal{N}(0,\varepsilon^2), \boldsymbol{W}\sim\mathcal{N}(0,\varepsilon^2),b\sim\mathcal{N}(0,\varepsilon^2)$ and $x_{u,v,\beta}^{\[1\]}$$ (i) $ $ =\[\sum\limits_{\alpha=1}^{C_{0}} $$ \(\sum\limits_{p=-\infty}^{+\infty} $$ \sum \limits_{q=-\infty}^{+\infty} $$ x_{u+p, v+q, \alpha}^{\[0\]} $$ \cdot W_{p,q,\alpha,\beta}^{[1]} \cdot \chi(p, q)\)\] $ $ +b_{\beta}^{[1]} $ $= \varepsilon $$  \bar x_{u,v,\beta}^{\[1\]}(i)$$ , a =\varepsilon\bar{a}.$ Thus $\sigma(x_{u,v,\beta}^{[1]}(i)) $$ = \sigma(\varepsilon  \bar x_{u,v,\beta}^{[1]}(i)) $$ = \sigma(0) + \varepsilon $$ \bar x_{u,v,\beta}^{[1]}(i)\sigma'(0)+\varepsilon$ by taylor expansion. As a result, $\frac{d \bar a_{u,v,\beta}}{d t} $$ \approx\frac{1}{n} \sum\limits_{i=1}^n y_i \bar{\boldsymbol{x}}^{[1]}_{u,v,\beta}$, under assumption $\sigma(0)=0$ and $\sigma'(0)=1$ (Tanh satisfy this assumption).
>
> $\textbf{Point 9: }$
>
> Why is the supremum necessary in Theorem 1? Is there a chance that condensation stops again before $T_{eff}$?
>
> $\textbf{Reply 9: }$
>
> The supremum is necessary since the limiting procedure is not true for arbitrary time $t$. For instance, if we take $t=0$, the first limitation will be $0$, and the second limitation will not be $1$ since the orientation at initialization will not condense at two opposite directions. $T_{eff}$ is the time when our initial model is effective, which means the model will condense at some point between $0$ and $T_{eff}$. We remark that  condensation    never stops  before  $T_{eff}$.
>
> $\textbf{Point 10: }$
>
> Do the assumptions correspond to the condensation regime from (Luo et al., 2021)? If not, what do the assumptions stand for?
>
> $\textbf{Reply 10: }$
>
> Assumption 3 corresponds to the condensation regime as described by Luo et al. (2021), while assumption 1 aligns with the assumption proposed by Zhou et al. (2022). Assumption 2 assumes that the data can be bounded, which is feasible in real-world applications, and assumption 4 indicates the presence of a pair of opposing directions.
>
> $\textbf{Point 11: }$
>
> Does condensation also occur when the last layer is initialized with zeros?
>
> $\textbf{Reply 11: }$
>
> Our analysis suggests that condensation can indeed occur under these conditions, as the initialization with zeros still satisfies all the assumptions outlined in our manuscript.
>
> However, while this zero initialization meets the theoretical conditions for condensation, it is not a common practice in real-world neural network training. The primary reason for this is the homogeneity it introduces in the training dynamics of different parameters within the last layer, which potentially hinder the network's ability to diversify its learning paths and explore the parameter space effectively.
>
> $\textbf{Point 12: }$
>
> If the results apply when assumptions are violated, shouldn't it be possible to loosen the assumptions?
>
> $\textbf{Reply 12: }$
>
> Yes. We are still on working loosening assumptions but there are many technical details needed to be addressed.

---

> > ### Comment · Reviewer_FvGf · 2023-11-22
> > **Rebuttal acknowledgement**
> >
> > I would like to let the authors know that I will take into account their rebuttal for my final decision.
> >
> > Upon a quick read-through, the following questions still remain:
> >  - I see how $\Big|\sum\_{\beta} \big\langle \boldsymbol{a}\_\beta, \sigma\bigl(\boldsymbol{x}\_\beta^{[1]}(i)\bigr)\big\rangle\Big| \leq H\_1 W\_1 \sum\_{\beta=1}^{M} \\|\boldsymbol{a}\_\beta\\|\_\infty \\|\sigma(\boldsymbol{x}_\beta)\\|\_\infty.$
> >   However, with the inequalities  $\\|\boldsymbol{a}\_\beta\\|\_\infty \leq \sqrt{\ln M}$ and $\\|\sigma(\boldsymbol{x}\_\beta)\\|\_\infty \leq \sqrt{\ln M}$, which only hold with high probability(!), I would have expected $H_1 W_1 \sum\_{\beta=1}^{M} \\|\boldsymbol{a}\_\beta\\|\_\infty \\|\sigma(\boldsymbol{x}\_\beta)\\|\_\infty \leq H_1 W_1 M \ln M,$ which is not upper bounded by $M$. Furthermore, it should be stated explicitly that the inequality only holds with high probability.
> >  - The Taylor expansion is only a good approximation when $\boldsymbol{x}_\beta$ is close to zero. Is this captured by one of the assumptions or does this require an additional assumption?

---

> ### Author Response · Authors · 2023-11-23
> **Response to Reviewer FvGf**
>
> We agree with the reviewer that in the context by the reviewer, it is upper bounded by $\log M$. However, in our context, where $||\boldsymbol{a}_{\beta}|| _{\infty} \leq \varepsilon \sqrt{\log M}, ~~||\sigma\left(\boldsymbol{x} _{\beta}^{[1]}(i) \right)|| _{\infty} \leq \varepsilon \sqrt{\log M}, $ and, as $H_1, W_1$ are constants independent with $M$, then, by ignoring constants and as $M\rightarrow\infty$, we obtain immediately that $$M H_1 W_1 ||\boldsymbol{a}|| _{\infty}|| \sigma\left(\boldsymbol{x}^{[1]}(i)\right)|| _{\infty} \leq \varepsilon^2M \log M.$$
>
> The manuscript is revised accordingly.
> Since the scale of $\varepsilon$ is of order  $M^{-(1+\delta)/2}$ for some $\delta>0$ (based on our assumption of initial scale $\gamma>1$), then compared with $M^{-\delta}$, the magnitude of  $\log M$ can be ignored.
> The above relation can be guaranteed  by a basic calculus exercise: For any $\delta>0$, however small that is,  the following holds $$\lim_{M\to \infty} \frac{\log M}{M^\delta}\to 0,$$
> hence the estimate in our context reads
> $$\varepsilon^2M \log M=M^{-\delta}\log M\to 0. $$
> Therefore, we have $$  |{\sum_{\beta=1}^M \varepsilon^2 \left<\boldsymbol{a}_{\beta}, \sigma\left(\boldsymbol{x}_{\beta}^{[1]}(i)\right)\right> }|\leq M \log M \varepsilon^2\ll 1.$$

---

> > ### Author Response · Authors · 2023-11-23
> > **Response to Reviewer FvGf**
> >
> > For another question about Taylor expansion, we have $\boldsymbol{x} ^{[1]} _{\beta}:=\mathrm{vec} (\boldsymbol{x} ^{[1]} _{u,  v, \beta})$ on page 4 where $\boldsymbol{x} _{ u, v, \beta}^{[1]}:= \left[ \sum _{\alpha=1}^{C _{0}} \left( \sum _{p=-\infty} ^{+\infty} \sum _{q=-\infty} ^{+\infty} \boldsymbol{x} ^{[0]} _{u+p, v+q, \alpha} \cdot \boldsymbol{W} _{p,q,\alpha,\beta}^{[1]} \cdot \chi(p, q) \right) \right]+\boldsymbol{b} _{\beta} ^{[1]}$. Thus, in our settings, $\boldsymbol{W} \sim \mathcal{N} (0,\varepsilon^2),b\sim \mathcal{N}(0,\varepsilon^2)$ and with assumption 2 that $\boldsymbol{x} ^{[0]}$ is bounded, we have $\boldsymbol{x} _{\beta}$ is close to zero.

---

### Official Review · Reviewer_P4L3 · 2023-10-31

**Soundness:** 3 good
**Presentation:** 3 good
**Contribution:** 3 good
**Rating:** 6
**Confidence:** 3

**Summary:**

This paper studies the initial condensation phenomenon of training CNNs, supported by both experimental results and theoretical analysis.

**Strengths:**

Understanding the training dynamics of gradient-based method is a crucial theoretical issue. While previous research has primarily concentrated on fully connected networks (FCNs), this paper represents a significant advancement in comprehending the training dynamics of CNNs. It investigates the initial condensation dynamics in CNNs, supported by comprehensive experimental evidence. Furthermore, the authors provide a precise mathematical characterization and time estimation for this initial condensation phenomenon in CNN training.

**Weaknesses:**

Assumption 4, while somewhat strict, effectively explains the initial condensation phenomenon. As discussed in my first quenstion below, I think that by relaxing this assumption, we can gain a more profound insight into the condensation phenomenon.

**Questions:**

- If Assumption 4 becomes $\lambda_1=\lambda_2>\lambda_3$, how will the initial condensation phenomenon change? My guess is that there will be two condensation directions, corresponding to the first two eigendirections. Is that right?

- In the initial stage of training, if we decompose the dynamics in polar coordinates, the radial velocity is substantially smaller than the tangential velocity. Does the time estimation provided in Theorem 1 relate to this property of the training dynamics?

---

> ### Author Response · Authors · 2023-11-18
> **Response to Reviewer P4L3**
>
> $\textbf{Point 1: }$
>
> If Assumption 4 becomes $\lambda_1=\lambda_2>\lambda_3$, how will the initial condensation phenomenon change? My guess is that there will be two condensation directions, corresponding to the first two eigendirections. Is that right?
>
> $\textbf{Reply 1: }$
>
> Yes this will make the situation extremely complicated. Since solving the linear dynamics involves diagonalization of the matrix, it could happen that the matrix would decompose into a Jordan form.  But it is very rare to have such case of $\lambda_1=\lambda_2>\lambda_3$.
>
> $\textbf{Point 2: }$
>
> In the initial stage of training, if we decompose the dynamics in polar coordinates, the radial velocity is substantially smaller than the tangential velocity. Does the time estimation provided in Theorem 1 relate to this property of the training dynamics?
>
> $\textbf{Reply 2: }$
>
> The radial velocity is much smaller than the tangential velocity, and the time estimated in Theorem
> 1 is not related to this property. Our linear approximate holds true throughout the time estimated in Theorem 1, which is the time required for $\theta_W$ to grow from $M^0$ to $M^{\delta}$, i.e.,$\theta_W(t)=t\exp(\lambda_1 t)\theta_W(0)$, thus we have $M^{\delta}= \exp(\lambda_1 t)M^{0}$, then $t\sim O(\log M)$.

---

### Meta-Review · Area_Chair_yd2Z · 2023-12-17

**Metareview:**

This paper looks at condensation in convolutional neural networks from both a theoretical and empirical perspective. Reviewers found the question important, but they struggled to follow the paper due to challenges with the writing, and they were confused about the significance of the eventual results. I defer to the reviewers and suggest rejecting the paper.

**Justification For Why Not Higher Score:**

The reviewers struggled to follow the paper. Even the authors acknowledged that the reviewers were asking "basic questions." I think that was confusing writing rather than lack of knowledge on the part of the reviewers (although this is a pretty niche topic that I don't really understand either).

**Justification For Why Not Lower Score:**

N/A

---

### Decision · Program_Chairs · 2024-01-16

Reject